# Evaluating and Comparing the Usability of Mobile Banking Applications in Saudi Arabia

Sarah Alhejji [1,2], Abdulmohsen Albesher [1,*], Heider Wahsheh [1] and Abdulaziz Albarrak [1]

1 The Department of Information System, College of Computer Sciences and Information Technology, King Faisal University, Hofuf 31982, Saudi Arabia
2 The Department of Management Information Systems and Production Management, College of Business and Economics, Qassim University, Buraydah 52571, Saudi Arabia
* Correspondence: aalbesher@kfu.edu.sa

**Abstract:** In many countries, the rapid growth of the Internet and mobile technologies has led to the expansion of Internet banking, especially mobile banking. Many banks seek to provide integrated banking services through mobile applications (apps) to increase customer satisfaction and loyalty. A quick look at the reviews of the mobile banking apps in Saudi Arabia reveals different usability issues among these apps. This research analyzed, evaluated, and compared the usability of all Saudi mobile banking apps available for the iOS and Android systems. Usability (as defined by ISO 9241) was measured using three criteria—effectiveness, efficiency, and satisfaction. This research also identified and discussed the most critical weaknesses of the Saudi banks' apps in regard to providing satisfactory solutions to developers. The results showed that the most critical issues existed in the user interfaces and functionality of the apps, especially those that frequently received updates. Furthermore, the lack of customer support made the interaction between banks and customers weak, leading to customer dissatisfaction.

**Keywords:** usability; user experience; mobile banking; customer review

## 1. Introduction

Recently, the mobile app industry has boomed dramatically worldwide. The number of mobile app downloads increased from 140.68 billion in 2016 to 230 billion in 2022 [1]. Rizk stated that mobile apps allow users to better share their feelings and opinions about the delivered services through writing reviews [2]. Reviews often contain valuable information for developers, as they can help them enhance and develop their apps to meet users' needs and desires. Specifically, reviews normally include requests for improvement, users' evaluations, bug reports, queries, or general descriptions of user experiences that can be positive or negative.

Srisopha et al., showed that app ratings and reviews are essential factors that users consider when choosing apps to download [3]. Moreover, the more positive comments and reviews an app has, the higher it ranks in the search results, which increases the chances of appearing to potential users. Reviews can indicate customer satisfaction with the apps and the services provided. Therefore, studying an app's reviews is an important part of the app life cycle and one of the most critical activities required of app developers in order to maintain and improve their app.

A quick review of banking apps in Saudi Arabia showed that customers' experiences were generally not satisfactory. Customer feedback and comments indicated many problems with the banking services provided. Because of this, this research aimed to analyze, evaluate, and compare the usability of mobile banking apps belonging to several Saudi banks. Usability was measured by the criteria set by ISO 9241, which are effectiveness, efficiency, and satisfaction. Additionally, this research aimed to identify the most critical problems and weaknesses faced by Saudi banking apps in regard to providing satisfactory

solutions to developers. The results obtained were used to determine which apps had the best usability and the most critical problems and weaknesses, as well as the types of improvements that service providers should make to enhance mobile banking.

The rest of the paper is organized as follows. The Section 2 reviews the literature related to this research. The Section 3 includes detailed information about the research methodology used. The Section 4 provides an analysis of the data and a review of the research findings. Finally, the Section 5 reviews the main issues encountered with mobile banking in Saudi Arabia, followed by a set of recommendations for developers.

### 1.1. Mobile Banking Applications

Mobile apps are essential components of modern information and communication technology. They allow users to have quick and easy access to the products, services, information, and processes they need in real time. The global use of modern technologies has changed interactions between business owners and customers, contributing to increasing customer loyalty and satisfaction. The development of apps requires an understanding of the target users in terms of their requirements, goals, and ideas. Ali et al. stated that the app industry requires more developer attention to develop apps for ease of use and reliability [4]. Ghandi et al., noted that mobile app development requires continuous improvements to meet new technological needs, such as the design of user interfaces in different sizes to fit the screens of mobile devices [5]. Many businesses have taken advantage of mobile apps to meet the needs of users and thus increase customer satisfaction and loyalty, including mobile healthcare apps (m-health), mobile learning apps (m-learning), and mobile banking apps (m-banking).

Customers can use m-banking to pay bills, transfer money, manage accounts, inquire about bank information, and search for ATM locations [6]. M-banking improves customer time management through instant communication and access to information, as well as its ability to be used anywhere. Because of this, it helps improve the customers' quality of life and increase the efficiency of banks [7]. Good-quality m-banking services can also help to retain and attract customers [8]. Furthermore, improvements in the provision of m-banking services contribute to increasing the bank's market share, reducing the cost of failure, and lowering the costs of business and attracting new customers to the bank.

With the rapid advancement of mobile technology, many banking customers in Saudi Arabia find it easy to use m-banking to make many financial transactions. Al-Khalidi noted that m-banking is the fastest-growing channel for financial growth in Saudi Arabia [9]. Al-Khalidi also explained that the spread of m-banking services is expected to continue in Saudi Arabia as the Internet infrastructure is modernized, government projects are implemented, and banking transactions and payment networks are strengthened and developed [9].

### 1.2. Customer Reviews

A customer review is a comment regarding a product or service written by a consumer who has used or experienced it. These reviews and ratings are public, which means developers and other users can read and benefit from them. As customers search online for product information and compare product options, they often have access to dozens or hundreds of product reviews written by other customers. The reviews and ratings can play an essential role in making purchase decisions. Askalidis and Malthouse indicated that 30% of customers under 45 years of age write reviews for every purchase they make, and 86% of customers say that those reviews are essential to making purchase decisions [10]. According to Avant, a company needs to handle user suggestions and complaints to survive, stay in the market, and gain customer loyalty [11]. Customer loyalty is critical, as retaining existing customers is more expensive than acquiring new customers. According to Zhang and Mao, the hotel sector has experienced a decline in customer loyalty because reading online reviews has increased customer understanding and purchasing power [12]. Avant found that when hotels responded personally to customer complaints and requests, the guest retention rate was 85% or higher, while hotels that did not respond to customer

demands retained around 30% of their customer base [11]. Therefore, implementing a customer–company interaction strategy is an effective way to maintain customer loyalty.

### 1.3. Usability

According to the ISO 9241-11 standard (ISO, 2018), usability is defined as "the extent to which a system, product, or service can be used by specified users to achieve specified goals with effectiveness, efficiency, and satisfaction in a specified context of use." As is stated in this definition, three factors (effectiveness, efficiency, and satisfaction) can affect usability [13]. ISO 9241-11 defines these factors as follows:

Effectiveness: the accuracy and completeness with which users achieve specified goals.

Efficiency: the resources used in relation to the results achieved.

User satisfaction: the extent to which the user's physical, cognitive, and emotional responses that result from the use of a system, product, or service meet the user's needs and expectations.

Usability and its associated factors have a decisive and powerful impact on the success of any system, website, or mobile app. Alber et al. defined efficiency as a measure of effectiveness that results in the minimum waste of time and effort [14]. Groth and Haslwanter linked efficiency with time, wherein they clarified the importance of time in measuring users' efficiency during the performance of tasks [15].

Similarly, mobile effectiveness is a significant factor in m-banking apps. Alber et al. defined this factor as the extent to which a goal is reached without regard to the method and resources for optimal use [14]. Groth and Haslwanter also linked effectiveness with accomplishing a task; when users fail to complete a simple task, this may be strong evidence that a bug in the application needs to be fixed [15].

Satisfaction is also an important factor that influences mobile app use. Lee et al., defined satisfaction as a brief emotional response from a mobile phone user [16]. Ravendran stated that satisfied m-banking users are more likely to buy from their banks than dissatisfied users [17]. Customers who are satisfied with services come back and buy again, telling others about their experiences. In contrast, customers who are deeply dissatisfied with services leave, while customers who are poorly satisfied with services may not leave but may complain [18].

In general, usability is essential and is considered one of the most important features of apps and software. One of the main reasons for the failure of apps and software is the need for a system to achieve the set goals of the users and measure their satisfaction. For this reason, usability assessment has become an essential part of any app or software development [19]. There are different methods to assess usability, the most famous of which is the System Usability Scale (SUS) and Sentiment Analysis (SA). On the other hand, the User Experience Questionnaire (UEQ) is one of the most popular measures for comprehensive user experience measurement.

The SUS is one of the most widely used usability testing tools today. The SUS scale was designed as a quick and easy way to assess usability [20]. The SUS scale consists of a set of data, ten verified data, covering five negative and five positive aspects of the system. Participants are asked to record each of the five questions. SUS scores can be grouped into percentage ranges [21], or a grading system can be used.

As for the UEQ measurement, it serves as a means to evaluate and measure the entire user experience. The UEQ scale is based on six scales, such as attractiveness, perspicuity, dependability, efficiency, stimulation, dependability, and novelty [20]. Scales are evaluated using pairs of opposite adjectives to describe the system, with participants choosing their level of agreement with each. UEQ scores assess how well a system meets users' expectations.

### 1.4. Sentiment Analysis (SA)

SA is a process that relies on identifying, classifying, and mathematically processing textual data to obtain the opinions and perspectives of users regarding the topics, services, and products offered to them [22]. Devika et al., mentioned another definition of SA, a

technique used to extract positive and negative user opinions about products or services offered [23]. These data are represented as customer feedback stored within forums, blogs, and social media.

Several methods can be used to gain user feedback, ranging from human analysis to machine learning. SA determines the polarity of data, which is news or product reviews. There are multiple ways of expressing emotions, from the three most common levels of polarity, positive, neutral, and negative, to scales of polarity, which are set, for example, from −10 to +10.

Luo et al., reported that opinion words dominate emotion indicators, particularly in adjectives, adverbs, and verbs, for example, "I love this app; it is amazing!" Opinion words are also known as emotion words, polar words, opinion lexicon, or opinion-carrying words, which can be categorized into two types: positive words such as cool, elegant, excellent, and negative words such as terrible, disgusting, and poor [24].

The feature extraction step is depended on SA frequencies to learn about the polarity of different reviews and comments. Weight scores for sentiment features are used to determine the strength and polarity of each review and comment. All sentiment features/words used in this study are manually extracted from comments and reviews collected from Saudi bank applications. TF (term frequency) refers to the number of times a given term (comment/review) occurs and is repeated. Then, each sentiment feature/word is weighed manually. The tool relies on the frequency of positive and negative terms/features to determine the polarity of the reviews. A review is considered positive when the frequency of positive terms/features exceeds the frequency of negative terms/features in the same review. A review is considered negative when the frequency of negative terms/features exceeds that of positive terms/traits in the same review. Finally, the review tool is considered neutral if the frequency of positive features/terms in the review equals the frequency of negative features/terms. Scores in polarity dictionaries are used by the tool to determine the strength of each entry [25].

## 2. Related Work

Omotosho analyzed reviews written by users of m-banking apps in Nigeria to extract valuable insights regarding the sentiments and emotions expressed by the users. The study found that around 66% of the emotions expressed by users were associated with anticipation, joy, and trust, whereby the remaining 34% were related to fear, disgust, surprise, and anger [26]. Tabiaa and Madani used online user evaluations to analyze the voice of the customer and to construct a topic modeling approach based on that data. Security, services, quality, and the user interface were the most common topics observed [27].

Permana et al., conducted a sentiment analysis and m-banking app review topic detection in Indonesia to determine customer sentiment toward m-banking apps and to learn which aspects of the examined apps needed to be maintained and improved. The most frequent topics observed among the negative reviews were app login problems, OTP code delivery constraints, and network connections. On the other hand, simplicity, helpfulness, and ease-of-use were the most frequent topics among the positive reviews [28]. Oh and Kim proposed a text-mining approach to identify factors that improved customer satisfaction when using m-banking applications. Their study showed that positive responses regarding the security and convenience of m-banking apps improved the rating of apps in stores. In contrast, increasing comments about insecurity, negative customer support experiences, and sophistication correlated with lower user ratings. These results support the idea that security is the most influential factor in customer satisfaction with m-banking services [29].

Metlo et al., conducted an empirical analysis to study the effect of m-banking on customer satisfaction in the Pakistani banking sector. The results showed that ease of use, credibility, and customer attitude significantly influenced customer satisfaction with the banking services provided [30]. Mkpojiogu et al., studied demographic differences in user satisfaction with the usability of m-banking apps. The results showed significant differences in the satisfaction of m-banking users based on gender, age, educational qualifications, and

experience. These results are helpful for banks, as they can help improve the interfaces of m-banking apps to better meet users' needs [8]. Gomachab and Maseke studied the effects of m-banking on customer satisfaction in commercial banks in Namibia. The results showed that the most-used service provided by the apps was the ability to make airtime purchases, and the least-used service was the allocation of funds [31].

Kaya et al. measured the usability of mobile apps using the SUS. This study aims to reveal the usability difference between four commonly used mobile apps: WhatsApp, Facebook, YouTube, and Mail. The study also looks for the difference in usability between iOS and Android operating systems. In this study, the SUS with an Adjective Rating Scale was applied to 222 young participants using these apps on their mobile phones. The results showed that the usability of all apps is somewhat satisfactory and above the standards. In more detail, the results showed that the usability of WhatsApp is higher compared to other apps, while the Facebook app has the lowest score. In addition, according to the results, there is no difference in the usability of mobile apps between operating systems [32]. Kristanto et al., assessed the usability of Ruang Guru apps using a UEQ. This survey focuses on user satisfaction, measured using a questionnaire with the Don Foundation Standard. In this study, measurements made on 100 active users of the Ruang Guru apps using the questionnaire were used randomly. The results of the measurements showed a degree of effectiveness of 59.38, efficiency of 65.36, and a degree of satisfaction of 62.52 [33].

## 3. Materials and Methods

The text data examined in this study consisted of reviews submitted by users of m-banking apps provided by 11 Saudi banks (data are available in a publicly accessible repository). These banks were Alinma Bank, Riyad Bank, Al-Rajhi Bank, SABB Bank, Al-Ahly Bank, Al-Fransi Bank, Al-Jazira Bank, Al-Bilad Bank, Arab Bank, Samba Bank, and Investment Bank. This research applied sentiment analysis to analyze reviews written in English and Arabic. The main goal of using SA was to identify the polarity of users' views about different aspects of app usage [34]. Moreover, the SA method is distinguished from others in that it classifies, identifies, and analyzes a large set of data without the need to identify participants or create tools for analysis, such as questionnaires and personal interviews. It is also an open method that allows any user to express his/her opinion directly at any time. In contrast, both SUS and UEQ need to create specific evaluation tools, such as questionnaires directed to a specific number of participants and containing specific questions asked by the analysts. The schematic overview of our approach is exhibited in Figure 1.

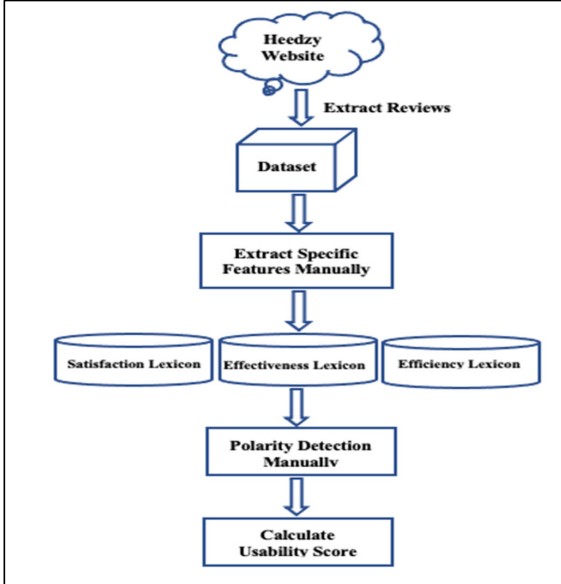

**Figure 1.** Review analysis schema.

The methods used for data collection and processing are explained below.

A.　The collection of data was performed using Heedzy, an online tool that allows for the downloading of mobile app reviews and ratings for Android and iOS. This study collected 5958 reviews from Android users and 2438 reviews from iOS users. The data were collected between January and March 2022. The next examples are shown from the collected reviews from the m-banking apps in Saudi Arabia.

Consider the following samples of the collected English reviews (original English review):
Example 1: The new update is bad, and the transaction process is slow and sometimes give errors.
Example 2: It is very good and quick

B.　Three polarity lexicons were constructed manually based on our selected usability factors—satisfaction, effectiveness, and efficiency. El-Halees built lexicons that used opinion phrases and words to determine the sentiment orientation of the whole review [19]. Our lexicon included 491 words describing satisfaction (278 positive, 213 negative), 265 describing effectiveness (11 positive, 254 negative), and 65 words describing efficiency (12 positive, 53 negative). Table 1 shows that the satisfaction factor contained more positive words than the effectiveness and efficiency factors. At the same time, satisfaction and effectiveness factors had negative words in close proportions and were much higher than the efficiency factor.

**Table 1.** Summary of polarity distribution among the usability factors.

| Usability Factors | No. of Positive Polarities | No. of Negative Polarities |
| --- | --- | --- |
| Satisfaction | 278 (65.5%) | 213 (43.3%) |
| Effectiveness | 11 (2.2%) | 254 (51.7%) |
| Efficiency | 12 (2.4%) | 53 (10.8%) |

Moreover, Tables 2–4 present a sample of positive and negative polarity words of the usability factors extracted from Saudi bank reviews. These features are stored in the lexicons for later use to determine the polarity of reviews.

**Table 2.** Satisfaction lexicon (English and Arabic words).

| Satisfaction Lexicon | | | |
| --- | --- | --- | --- |
| **Positive** | | **Negative** | |
| Arabic | English | Arabic | English |
| ممتاز | Excellent | سيء | Bad |
| جيد | Good | قديم | Old |

**Table 3.** Effectiveness lexicon (English and Arabic words).

| Effectiveness Lexicon | | | |
| --- | --- | --- | --- |
| **Positive** | | **Negative** | |
| Arabic | English | Arabic | English |
| فعال | Effective | لا يعمل | Not working |
| يعمل | Working perfectly. | خطأ | Error |

**Table 4.** Efficiency lexicon (English and Arabic words).

| Efficiency Lexicon | | | |
|---|---|---|---|
| **Positive** | | **Negative** | |
| Arabic | English | Arabic | English |
| يوفر وقت | quick | بطيء | Slow |
| سريع | Fast | يعلق | Hanging |

C.   The reviews of each bank were labeled manually based on usability factors. Groth and Haslwanter found three traits that measured usability in any review of mobile usability models: effectiveness, efficiency, and satisfaction [15]. A polarity score was given by the authors' judgment for each factor, a quantitative measure of the positive or negative interactions expressed in the reviews. The average polarity score ranged from −1 to +1, with negative values indicating negative opinions, values of zero indicating neutral opinions, and positive values indicating positive opinions.

D.   The total usability score was calculated for each review based on the sum of each review's satisfaction, effectiveness, and efficiency values [19]. The overall usability score ranged from −3 to +3. The usability of all banks was evaluated and compared based on these scores in order to determine which of the banks had the highest and lowest usability.

Based on example 1 (step A), Table 5 exhibits how to give a polarity score for each word in these reviews that match the lexicons built. Then, the total usability score was calculated.

**Table 5.** Manually extracted features with its polarity weight and calculated usability score (example 1).

| Review | Satisfaction | Effectiveness | Efficiency | Usability Score |
|---|---|---|---|---|
| The new update is bad, and the transaction process is slow and sometimes gives errors. | −1 | −1 | −1 | −3 |

## 4. Results and Discussion

After arranging the reviews into separate excel sheets for each operating system, we categorized and determined the degree of each usability factor (satisfaction, effectiveness, efficiency). The usability score for each Saudi bank was calculated by summing the percentage of positive and negative reviews for each usability factor. Tables 6 and 7 show the following data for Android and iOS, respectively: the name of the bank, the number of reviews, the percentage of positive and negative reviews for each usability factor, and the overall usability scores.

Table 6 shows the usability scores of all Saudi bank apps on the Android system based on the values of our selected usability factors. These are satisfaction, effectiveness, and efficiency (both positive and negative). The Alinma Bank app demonstrated the highest positive usability ratio, calculated at 0.81, followed by the Riyad Bank app with a score of 0.72, the Al-Rajhi Bank app with a score of 0.71, the SABB Bank app with a score of 0.69, and Al-Ahli Bank app with a score of 0.34. Meanwhile, the Investment Bank app possessed the highest negative usability ratio, calculated at −0.93, followed by the Samba Bank app with a score of −0.49, the ANB Bank app with a score of −0.34, the Al-Bilad Bank app with a score of −0.30, the Al-Jazira Bank app with a score of −0.28, and the Al-Fransi Bank app with a score of −0.25.

**Table 6.** Usability score (Android system).

| | Bank Name | Total Reviews | Satisfaction | | | | | Effectiveness | | | | | Efficiency | | | | | Usability Score |
|---|---|---|---|---|---|---|---|---|---|---|---|---|---|---|---|---|---|---|
| | | | No. of Negative Reviews | Score [4] | % [3] | Positive [2] | Negative [1] | No. of Negative Reviews | Score | % | Positive | Negative | No. of Negative Reviews | Score | % | Positive | Negative | |
| Android | Alinma Bank | 405 | 12 | 343 | 0.847 | 0.877 | −0.030 | 21 | −20 | −0.049 | 0.002 | −0.052 | 8 | 6 | 0.015 | 0.035 | −0.020 | 0.81 |
| | Riyad Bank | 962 | 48 | 744 | 0.773 | 0.823 | −0.050 | 77 | −69 | −0.072 | 0.008 | −0.080 | 19 | 13 | 0.014 | 0.033 | −0.020 | 0.72 |
| | Al-Rajhi Bank | 1240 | 43 | 980 | 0.790 | 0.825 | −0.035 | 63 | −63 | −0.051 | 0 | −0.051 | 50 | −40 | −0.032 | 0.008 | −0.040 | 0.71 |
| | SABB Bank | 964 | 60 | 708 | 0.734 | 0.797 | −0.062 | 79 | −77 | −0.080 | 0.002 | −0.082 | 16 | 31 | 0.032 | 0.049 | −0.017 | 0.69 |
| | Al-Ahli Bank | 1085 | 132 | 556 | 0.512 | 0.634 | −0.122 | 177 | −176 | −0.162 | 0.001 | −0.163 | 22 | −12 | −0.011 | 0.009 | −0.020 | 0.34 |
| | Al-Fransi Bank | 142 | 29 | 31 | 0.218 | 0.423 | −0.204 | 52 | −52 | −0.366 | 0 | −0.366 | 15 | −14 | −0.099 | 0.007 | −0.106 | −0.25 |
| | Al-Jazira Bank | 326 | 88 | 56 | 0.172 | 0.442 | −0.270 | 102 | −102 | −0.313 | 0 | −0.313 | 49 | −45 | −0.138 | 0.012 | −0.150 | −0.28 |
| | Al-Bilad Bank | 305 | 58 | 63 | 0.207 | 0.397 | −0.190 | 123 | −122 | −0.400 | 0.003 | −0.403 | 34 | −33 | −0.108 | 0.003 | −0.111 | −0.30 |
| | ANB Bank | 353 | 97 | 47 | 0.133 | 0.408 | −0.275 | 155 | −155 | −0.439 | 0 | −0.439 | 19 | −12 | −0.034 | 0.020 | −0.054 | −0.34 |
| | Samba Bank | 105 | 47 | −16 | −0.152 | 0.295 | −0.448 | 26 | −26 | −0.248 | 0 | −0.248 | 9 | −9 | −0.086 | 0 | −0.086 | −0.49 |
| | Investment Bank | 71 | 33 | −21 | −0.296 | 0.169 | −0.465 | 26 | −25 | −0.352 | 0.014 | −0.366 | 20 | −20 | −0.282 | 0 | −0.282 | 0.93 |

[1] Negative: The number of negative reviews divided by the total number of reviews. [2] Positive: The number of positive reviews divided by the total number of reviews. [3] %: The score is divided by the total number of reviews. [4] Score: The number of positive reviews minus the number of negative reviews.

**Table 7.** Usability score (iOS system).

| | Bank Name | Total Reviews | Satisfaction | | | | | Effectiveness | | | | | Efficiency | | | | | Usability Score |
| --- | --- | --- | --- | --- | --- | --- | --- | --- | --- | --- | --- | --- | --- | --- | --- | --- | --- | --- |
| | | | No. of Negative Reviews | Score | % | Positive | Negative | No. of Negative Reviews | Score | % | Positive | Negative | No. of Negative Reviews | Score | % | Positive | Negative | |
| iOS | Al-Rajhi Bank | 668 | 26 | 514 | 0.769 | 0.808 | −0.039 | 35 | −33 | −0.049 | 0.003 | −0.052 | 9 | 33 | 0.049 | 0.063 | −0.013 | 0.77 |
| | Alinma Bank | 183 | 40 | 52 | 0.284 | 0.503 | −0.219 | 38 | −37 | −0.202 | 0.005 | −0.208 | 18 | −9 | −0.049 | 0.049 | −0.098 | 0.03 |
| | Riyad Bank | 429 | 106 | 75 | 0.175 | 0.422 | −0.247 | 122 | −122 | −0.284 | 0 | −0.284 | 64 | −52 | −0.121 | 0.028 | −0.149 | −0.23 |
| | SABB Bank | 129 | 30 | 16 | 0.124 | 0.357 | −0.233 | 50 | −50 | −0.388 | 0 | −0.388 | 15 | −11 | −0.085 | 0.031 | −0.116 | −0.35 |
| | ANB Bank | 156 | 53 | −23 | −0.147 | 0.192 | −0.340 | 87 | −87 | −0.558 | 0 | −0.558 | 15 | −12 | −0.077 | 0.019 | −0.096 | −0.78 |
| | Al-Bilad Bank | 81 | 30 | −21 | −0.259 | 0.111 | −0.370 | 45 | −45 | −0.556 | 0 | −0.556 | 12 | −10 | −0.123 | 0.025 | −0.148 | −0.94 |
| | Investment Bank | 86 | 35 | −29 | −0.337 | 0.070 | −0.407 | 27 | −26 | −0.302 | 0.012 | −0.314 | 31 | −31 | −0.360 | 0 | −0.360 | −1.00 |
| | Al-Jazira Bank | 155 | 41 | −30 | −0.194 | 0.071 | −0.265 | 101 | −101 | −0.652 | 0 | −0.652 | 40 | −39 | −0.252 | 0.006 | −0.258 | −1.10 |
| | Samba Bank | 48 | 37 | −36 | −0.750 | 0.021 | −0.771 | 16 | −16 | −0.333 | 0 | −0.333 | 2 | −2 | −0.042 | 0 | −0.042 | −1.13 |
| | Al-Ahli Bank | 466 | 253 | −237 | −0.509 | 0.034 | −0.543 | 229 | −229 | −0.491 | 0 | −0.491 | 148 | −147 | −0.315 | 0.002 | −0.318 | −1.32 |
| | Al-Fransi Bank | 37 | 25 | −24 | −0.649 | 0.027 | −0.676 | 21 | −21 | −0.568 | 0 | −0.568 | 6 | −6 | −0.162 | 0 | −0.162 | −1.38 |

Table 7 shows the usability scores for all Saudi banks in the iOS system. The Al-Rajhi Bank app demonstrated the highest positive usability ratio, calculated at 0.77, followed by the Alinma Bank app with a score of 0.03. Meanwhile, the Al-Fransi Bank app had the highest negative usability ratio, calculated at −1.38, followed by the Al-Ahli Bank app with a score of −1.32, the Samba Bank app with a score of −1.13, the Al-Jazira Bank app with a score of −1.10, the Investment Bank app with a score of −1.00, the Al-Bilad Bank app with a score of −0.94, the ANB Bank app with a score of −0.78, the SABB Bank app with a score of −0.35, and the Riyad Bank app with a score of −0.23.

Tables 6 and 7 show that the satisfaction factor plays an essential role in determining the degree of usability. This was followed by the factors of effectiveness and finally efficiency. Among Android users, the Alinma Bank app had the highest satisfaction rate among banking applications, calculated at 0.877, followed by the Al Rajhi Bank app with a score of 0.825, and then the Riyad Bank app with a score of 0.823. Meanwhile, the Investment Bank app had the lowest satisfaction rate among all bank applications, with a calculated rate of −0.465, followed by the Samba Bank app with a score of −0.448, and the ANB Bank app with a score of −0.275. Among iOS users, the Al-Rajhi Bank app had the highest satisfaction rate, calculated at 0.808, followed by the Alinma Bank app with a score of 0.503, and then the Riyad Bank app with a score of 0.422. Meanwhile, the Samba Bank app had the lowest satisfaction rate, calculated at −0.771, followed by the Al-Fransi Bank app with a score of −0.676, and then the Al-Ahli Bank app with a score of −0.543.

From Tables 6 and 7, we can see that opinions regarding effectiveness were weighted toward negative in reviews of all Saudi bank applications. Among Android users, the ANB Bank app had the lowest effectiveness rate among all Saudi bank applications, with an estimated rate of −0.439, followed by the Al-Bilad Bank app with a score of −0.403, and then the Al-Fransi and Investment Bank apps, both with scores of −0.366. Among iOS users, the Al-Jazira Bank app possessed the lowest effectiveness rate, with a calculated value of −0.652, followed by the Al-Fransi Bank app with a score of −0.568 and the ANB Bank app with a score of −0.558.

In addition, Tables 6 and 7 revealed that efficiency scores were often close to the neutral zero. Among Android users, the SABB Bank app possessed the highest efficiency ratio of the banking applications, with a calculated percentage of 0.049, followed by the Alinma Bank app with a score of 0.035, the Riyad Bank app with a score of 0.033, and the ANB Bank app with a score of 0.020. Meanwhile, the Investment Bank app possessed the lowest efficiency rate of the banking applications, with a calculated percentage of −0.282, followed by the Al-Jazira Bank app with a score of −0.150, the Al-Bilad Bank app with a score of −0.111, and the Fransi Bank app with a score of −0.106. Among iOS users, only one bank, the Al-Rajhi Bank, possessed a high efficiency rate, with a score of 0.063. The Investment Bank app once again had the lowest efficiency rate, this time calculated at −0.360, followed by the Al-Ahli Bank app with a score of −0.318 and the Al-Jazira Bank app with a score of −0.258.

Our research indicated various issues that affected usability from the perspectives of customer satisfaction, effectiveness, and efficiency. Our quantitative analysis led us to find common patterns that the banks have regarding these aspects and thus complement the results obtained from the usability evaluation. We found that these issues mainly stemmed from new updates, disabled functions, and lack of customer support.

New Update Problem

One of the commonly repeated problems that affected customers' satisfaction with the Saudi m-banking apps on both the Android and iOS was the issue of new app updates. For example, the comment "problem with the new update" appeared frequently in reviews of the Al-Jazira Bank app, comprising an estimated 28.5% of reviews on Android and 24.5% on iOS. In regard to the Investment Bank app, the problem appeared with a frequency of 11% on Android and 11.7% on iOS. Meanwhile, 9.6% of reviews regarding the ANB Bank app on iOS referred to the new update issue using the repeated comment "this app does not work on a jailbreak device," despite the evidence that their device was jailbreak-free.

Update issues have also been found in apps in other domains, such as health care. For example, Alqahtani and Orji mentioned that users of some mental health apps explained that a new update caused some problems with the app's functions that led to data loss [35]. In the m-banking domain, Tabiaa and Madani also reported frequent issues with new updates [27].

Functional Problems

The main objective of using m-banking apps is to complete banking transactions in a quick and easy manner. Baabdullah et al., mentioned that Saudi customers view m-banking as a method that saves customers time, money, and effort [36]. A strong relationship exists between the actual use of banking services and customer loyalty [36]. They indicated that the implementation of m-banking functions is essential not only for customer loyalty but also for increasing customer satisfaction [36].

In regard to the effectiveness factor, the apps reviewed in this research showed that customers complained about the disabling of some app functions by some Saudi banks. Commonly repeated comments regarding this issue included "the application does not work" representing 15% of comments, "cannot log in" 13% of comments, "the application cannot be opened" 11% of comments, and "there is a general error" 8% of comments. The ANB Bank apps for both Android and iOS systems, the Al-Bilad Bank and the Investment Bank apps for the Android system, and the Al-Jazira Bank and the Al-Fransi Bank apps for the iOS system were ranked as the worst apps regarding this issue. Similar issues were found in other research papers. For example, Permana et al., conducted a study on m-banking apps in Indonesia; their study showed that the most common issues involved signing in to an app [28].

Lack of Customer Service Support

Saudi banking apps should provide different methods to communicate with customers. One possible method is via an instant chat, which only the SABB Bank app provided. Meanwhile, the apps belonging to the Al-Rajhi, Al-Fransi, and Alahli banks did not provide an instant chat, but they did allow customers to send their queries and complaints through the apps. Many reviews of the Saudi banking apps contained questions and inquiries from customers directed to the app developers. These reviews, which represent 7% of all comments, include a range of in-app complaints and questions about how to use the app, the contents of new updates, and how to activate certain services. Some reviews that included a specific complaint about the app were answered by app developers with short responses—for example, "Thank you for your note; the problem will be resolved as soon as possible." Neglecting to respond to customers can negatively affect customer satisfaction. Banks should respond to customers swiftly and direct them to proper solutions. Sharma and Sharma emphasized the need for a well-trained staff who can listen to, understand, and handle customers' problems [37].

Based on our analysis of users' positive and negative reviews in Saudi banks apps, problems and themes were extracted from our results. We recommend the following to developers in order to improve the usability of Saudi banking apps:

- Examine new updates before they are officially released and ensure that they are free from errors and problems. Many reviews indicated that customers were not satisfied with banking apps due to issues with new updates, which caused issues such as applications that stopped working and updates that were not compatible with the user's mobile device.
- Analyze app reviews to improve apps and keep customers satisfied. Reviews are shared spaces that allow customers to express their opinions, requests, and complaints regarding downloaded apps. These reviews allow developers to receive customer feedback about app usability issues. If these reviews are taken into account and the developers fix the problems, it will significantly improve the apps.
- Respond quickly to app reviews to increase customer satisfaction. Developers' responses to customer comments increase customer satisfaction and loyalty. These responses make the customer feel that their feedback is important, and that the developer is keen to solve

their problems. Furthermore, responding to reviews has positive results—ratings of the app often increase after developers respond to customer reviews.

- Enable a live chat function that can support interaction between banks and their customers. This would allow for an effective channel of communication that could respond to user requests and fix recurring issues in real time.

## 5. Conclusions

Customer feedback and comments on m-banking apps belonging to some Saudi banks indicate many issues in the banking services provided. This study aimed to evaluate and compare the usability of all Saudi m-banking apps for Android and iOS based on three usability factors: satisfaction, effectiveness, and efficiency. This research identified several usability issues and recommended some useful solutions. Saudi banks should examine and review new updates before they are issued, ensure that the essential functions of the apps work, and consider adding online chat features for customer service. The difficulties this study encountered are represented in the study and analysis of reviews in the Arabic language, as the Arabic language is considered difficult because it is a highly inflectional and derived language. Moreover, some comments include words in the Saudi dialect, which are somewhat inconsistent with the Arabic language, in addition to the fact that these words do not have a particular lexicon. Additionally, some confusing words were excluded from this study, including positive words in some regions of Saudi Arabia and negative ones in others. Some comments include positive words and negative symbols that are difficult to classify; thus, they were excluded.

The iOS and Android app stores recently began allowing developers to respond to customer reviews. As part of our future work, we plan to study and analyze the responses of app developers to customer reviews, evaluate the quality of those responses, and determine whether the developer's response affects the customer's experience. Moreover, a model can be created for this work so that reviews are categorized and analyzed automatically instead of by the manual method used.

**Author Contributions:** Data curation, S.A.; Formal analysis, S.A.; Methodology, S.A.; Supervision, A.A. (Abdulmohsen Albesher), H.W. and A.A. (Abdulaziz Albarrak); Writing—original draft, S.A.; Writing–review and editing, A.A. (Abdulmohsen Albesher) and H.W. All authors have read and agreed to the published version of the manuscript.

**Funding:** This work was supported by the Deanship of Scientific Research, Vice Presidency for Graduate Studies and Scientific Research, King Faisal University, Saudi Arabia (grant no. 1074), through its KFU Research Summer initiative.

**Data Availability Statement:** Data available in a publicly accessible repository. These data can be found here: (https://drive.google.com/drive/folders/1ZdG5efrKos0p0-hHT8BsO3uKvC4AOdnu accessed on 9 October 2022).

**Conflicts of Interest:** The authors declare no conflict of interest.

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
