# Peer review of "Evaluating and Comparing the Usability of Mobile Banking Applications in Saudi Arabia"

_information, doi:10.3390/info13120559_

Round 1

Reviewer 1 Report

The work is an interesting proposal concerning the evaluation of usability in software. The context of application of the study is of interest since new banking applications are generated every day.

The work addresses in a minimal or little way the different usability evaluation techniques that could contrast well with the proposed ones. Sentiment analysis is also mentioned, which is also only lightly addressed in this proposal.

In section 3. It is recommended to create a map of the lexicons in item B to present the description of the proposed lexicons.

What other mechanisms were considered for the evaluation of usability? Although there are existing instruments such as the UEQ, what is the contribution regarding the existing ones?

Even in the related work, it would be interesting to discuss the various usability evaluation mechanisms existing in the literature.

The usability tables 1 and 2 could be improved by showing the total number of negative reviews in order to know the number for each factor. 

4. Results and Discussion

In the results and discussion section, other problems encountered such as the app interface, updates, disabled features, and lack of customer support are mentioned. 

In this sense, the study can be used to present quantitatively the patterns that the banks have in common regarding these aspects and thus complement the results obtained from the usability evaluation.

The conclusions section can be improved by pointing out the problems encountered and the challenges, as well as the future work to be done on the proposal.

Reviewer 2 Report

Thanks for taking the time to write this article. I consider the topic and the approach interesting. The main approach is new to me.

The approach is described in section 3.

I understand step A.

Step B says, “Three polarity lexicons were constructed ...” This is key to understanding the research. How was this done and by whom? If you used the lexicon created by El-Halees, how did you verify that it works for banking apps?

Please provide some examples of phrases and words from your polarity lexicons and specify whether they are positive, neutral or negative, and whether they pertain to effectiveness, efficiency or satisfaction.

Step C says, “The reviews of each bank were labelled based on usability factors.” How was this labelling done and by whom? The same questions apply to your statement, “A polarity score was given for each factor ...”

I think that I understand step D, but I miss a literature reference that confirms that it’s valid to add the scores for effectiveness, efficiency and satisfaction to obtain a one-figure usability score.

I also have some comments on the total approach, step A to D:

·        What is a “polarity score”? This concept is so important for your approach that I think an explicit explanation with examples in the paper is required.

·        How did you compare the scores from the automated analysis in step B and C to the scores given by a neutral human being? I am not asking for a complete scoring of all 5,958 + 2,438 reviews by a human being, but a scoring of a random sample of, say, 5% of the reviews.

·        I miss examples of real feedback from users and the corresponding scores for effectiveness, efficiency and satisfaction so the readers of the article and I can judge for ourselves how reliable the automated analysis is.

My advice regarding the description of step A to D: Add a graphic that explains your approach, and avoid the passive form, so it becomes clear who acts.

In the section, “User Interface Problems”, the paper turns to a different topic, which is not covered by the title. The section reports five usability problems. Descriptions of usability problems must be clear and they must be accompanied by the number of users who reported each problem. I understand the description of four of the problems, but I fail to understand the meaning of “... and did not contain a sequence for the services”

I agree with the authors that “Functional Problems” and “Lack of Customer Service Support” could be important usability issues for the banks. However, these sections are entirely qualitative. I encourage the authors to report specific figures to substantiate their claims.

I think a header is missing before “Based on our results, we recommend the following ...”

I looked up the app for alrajhi bank on Google Play. Google Play says that there are 933k reviews and that the average score is 4.5 out of 5. This score doesn’t sound too bad to me and these figures should be mentioned and discussed in the paper. Please explain the discrepancy between the “933k” reviews reported by Google Play and the 1,240 reviews reported in Table 1 in the article. Please consider adding Google’s figures to Table 1 or 2.

The current top-3 reviews match the finding in the section, “Functional Problems”, quite well.

I miss the section, “Discussion”, where the authors critically reflect on their findings and their validity. In particular, I miss a discussion of the basic question: Are these reviews representative? For example, are the 1240+668 reviews of the app from the Al-Rajhi Bank just the tip of the iceberg? Or does this app have, say, 2,000,000 active users who are entirely happy with the app? Google says that the number of downloads of this app is “5M+”

Round 2

Reviewer 1 Report

I would like to thank the authors for their timely attention to the comments made.

 On my part, the authors have responded to the comments made.

Author Response

Thank you for your comment, and we appreciate your work. 

Reviewer 2 Report

The paper has improved considerably. I now better understand the approach the authors have used.

The article has apparently been improved by adding new text without deleting any text. The first version of the paper was 12 pages. The revised version is 19 pages, which is not justified by the scientific value of the paper. I strongly recommend that the paper is shortened by at least five pages. Suggestions:

- Shorten or remove section 1.3 and 1.4. In particular, carefully consider the need for each of the additions you made. I recommend that all additions are removed.

- Remove the additions to section 2

- Merge example 1, 2, 3 and 4 into one example

- Shorten the list of references to two pages

- Remove the added text in “Other Problems” on page 14. This is purely speculative and not supported by data or user research

I further recommend the following changes:

- Increase the size of Figure 1 so it becomes easier to read.

- Edit Table 7 so it appears on one page. Tables should never be divided over several pages

- Edit Table 8 so the caption appears on the same page as the table.

I appreciate that the authors made the raw data available (Data Availability Statement). I found the raw data quite helpful in understanding the approach and the results. I recommend that the authors mention this explicitly in the paper and not just in an inconspicuous note at the end of the paper
